# Population genetic analysis of *Aedes aegypti* reveals evidence of emerging admixture populations in coastal Kenya

Francis Mulwa[1,2]*, Dario Balcazar[3], Solomon Langat[1], James Mutisya[1], Betty Chelangat[1], Carolyn S. McBride[4], Noah Rose[4], Jeffrey Powell[5], Rosemary Sang[1], Armanda Bastos[2,6], Andrea Gloria-Soria[3], Joel Lutomiah[1]

**1** Center for Virus Research, Kenya Medical Research Institute, Nairobi, Kenya, **2** Department of Entomology, The Connecticut Agricultural Experiment Station, New Haven, Connecticut, United States of America, **3** Department of Ecology and Evolutionary Biology, University of Princeton, Princeton, New Jersey, United States of America, **4** Ecology and Evolutionary Biology Department, Yale University, New Haven, Connecticut, United States of America, **5** Department of Zoology & Entomology, University of Pretoria, Pretoria, South Africa, **6** Department of Veterinary Tropical Diseases, University of Pretoria, Pretoria, South Africa

* fmusili85@gmail.com

## Abstract

### Background

The *Aedes aegypti* mosquito is widespread in tropical and subtropical regions. There are two recognized subspecies; the invasive *Aedes aegypti aegypti* (*Aaa*) and the ancestral *Aedes aegypti formosus* (*Aaf*). *Aaf* is common throughout Kenya whereas *Aaa*, which was historically confined to coastal regions, has undergone a range expansion. In areas of sympatry, gene flow may lead to admixed populations with potential differences in vectorial capacity. We hypothesize that coastal *Ae. aegypti* populations have a higher proportion of *Aaa* ancestry than those from inland locations of Kenya, influenced by their distance to the coast.

### Methodology

Adult *Ae. aegypti* mosquitoes were collected using Biogent (BG) sentinel traps baited with carbon-dioxide ($CO_2$) from cities and towns along the Kenyan northern transport corridor. *Aedes aegypti* population structure, genetic diversity, and isolation by distance were analyzed using genome-wide single nucleotide polymorphism (SNPs) datasets generated with an *Ae. aegypti* microarray chip targeting ≈50,000 SNPs. Kenyan *Aedes aegypti* populations were placed into a global context within a phylogenetic tree, by combining the Kenyan dataset with a previously published global database.

**Data availability statement:** The datasets generated and analysed in this study are available in the supplementary material and in the manuscript. Additional raw data and output files are available in Github through the following link: https://github.com/fmusili85/SNPs-Pop-Ae.-aegypti-Kenya.

**Funding:** This study was funded by the Kenya government through the National Research Fund (NRF) Kenya, Grant number NRF/1/MMC/28. The funders played no role in the study design, data collection, analysis or manuscript preparation.

**Competing interests:** The authors have declared that no competing interests exist

## Results

A total of 67 *Ae. aegypti* mosquitoes population from Kenya were genotyped, we found that western Kenya *Ae. aegypti* constitute a genetically homogenous population that clusters with African *Aaf*, whereas coastal mosquitoes showed evidence of admixture between the two subspecies. There was a positive correlation (Observation = 0.869, p = 0.0023) between genetic distance (FST) and geographic distance, suggesting isolation by distance. The phylogenetic analysis and the genetic structure analysis suggest that an Asian Aaa population is the source of Aaa invasion into Kenya.

## Conclusions

These results provide evidence of an emerging admixed population of *Ae. aegypti* in coastal Kenya between the sylvatic *Aaf* and the domesticated-human preferring *Aaa*. The observed gene flow from *Aaa* into Kenya may positively influence *Ae. aegypti* vectorial capacity, potentially increasing human feeding preference, biting rates and vector competence and could be promoting the observed dengue and chikungunya outbreaks.

## Author's summary

*Aedes aegypti* is of great public health concern due the viruses they transmit. The vector is highly invasive and is expanding to new geographic regions, quickly adapting to new environment. This study focuses in understanding the genetic structure of *Ae. aegypti* in cities along the northern transport corridor in Kenya, which are at risk of vector invasion. We analyzed *Ae. aegypti* populations using a panel of Single Nucleotide Polymorphism (SNPs) markers distributed across the genome. Our analysis shows admixture population in the coastal region between *Aaf* and *Aaa*, with Asian mosquitos being the putative source of *Aaa* ancestry, while the western populations are more related to African *Aaf*. This research provides a broad picture of the nature and dynamics of the *Ae. aegypti* populations across Kenya and sets the bases for further genetic studies focused on improving vector control strategies and developing novel mosquito control methods.

## Introduction

*Aedes aegypti* is the primary vector for arboviruses such as dengue, chikungunya, Zika, and yellow fever viruses, which cause frequent outbreaks mostly in tropical and subtropical regions across the world [1]. This species has two widely recognized subspecies: *Aedes aegypti formosus* (*Aaf*), the sylvatic type that dominates across most of sub-Saharan Africa, and *Aedes aegypti aegypti* (*Aaa*), the invasive type of notable epidemiological relevance due to its anthropophagic traits [2–5]. *Aaa* is believed to have originated in the Sahel region of West Africa but has since successfully invaded

the tropics around the world, and is increasingly reported in temperate regions where it is associated with emerging out-breaks of Zika, dengue and yellow fever [6]. Human activities such as unplanned urban expansion, population movement, and worldwide trade have contributed to the adaptation of this invasive species to the urban environment, and its association with native species [7–10]. *Aedes*-borne arboviral infections have become a major public health concern in Kenya since 2004/2005 chikungunya outbreak, and are rapidly spreading to new geographic regions [11–14]. Entomological investigations during risk assessment and routine surveillance activities have shown that *Ae. aegypti* is distributed across the country, but predominant in coastal Kenya [15–17].

*Ae. aegypti* micro geographic population structure in Africa has revealed the presence of admixed populations of the two subspecies [10, 18, 19]. In Kenya, previous genetic analyses indicated that the *Ae. aegypti* found inside residential houses in the coastal region are descendants of the invasive subspecies *Aaa* introduced to the east coast of Kenya [18, 20]. These foreign mosquitoes have coexisted with the native *Aaf* for at least 60 years and were last recorded in 2009 [21]. The two forms share same habitat and breeding containers both indoors and outdoors and it is likely that interbreeding has occurred [22–24]. A recent study reported that *Ae. aegypti* population in coastal Kenya is a hybrid of the two subspecies [22, 25]. Thus the overlap of the two subspecies could lead to admixture, as observed in other parts of Africa [26, 27]. As a result of gene flow from *Aaa* into Kenya, the resulting ecotype may exhibit enhanced vectorial capacity, such as high human attack rates and increased vector competence which could increase transmission of diseases [25, 28]. Although some admixture between the two subspecies has been shown in Kenya [18], the extent of admixture across mainland Kenya is still largely unknown.

Genome-wide analysis using single nucleotide polymorphisms (SNPs), has proven powerful for high-resolution analysis of *Ae. aegypti* evolutionary history and invasion dynamics [22, 29, 30]. Gaining a better understanding of the genetic structure of *Ae. aegypti* in Kenya, along with inferring *Aaa* invasion dynamics and levels of *Aaa* gene flow into the country, can provide valuable insights for controlling and forecasting future disease outbreaks. The Kenyan northern transport corridor is the main passageway connecting several countries in Eastern Africa. It is comprised by the meter gauge railway line and a highway that ferry passengers and cargo from Mombasa to neighboring border countries such as Uganda, Tanzania and South Sudan. Mombasa city is an international tourism hub, as well as a logistics and communications hub for neighboring countries, making it highly susceptible to invasion by foreign species. Initial invasion followed by human-mediated passive dispersal is likely to facilitate spread to other urban areas in mainland Kenya.

We performed population genetic analyses on *Ae. aegypti* collected from diverse ecological zones along a corridor from coastal Kenya to the mainland regions to better understand the *Aaa* admixture across the country. A SNP genotyping array which targets 50,000 SNPs [31] was employed to investigate the population structure of *Ae. aegypti* in Kenya and determine the genetic diversity and differentiation in the urban cities along the Kenyan northern transport corridor. Such knowledge allows inference of the invasion process and underpins modelling of the spread of disease and insecticide resistance. Additionally, it is useful for maintaining effective vector control strategies and provides insight into the epidemiology of arthropod-borne diseases such as dengue virus.

## Methods

### Ethical approvals

Ethical approval for this study was obtained from Kenya Medical Research Institute's (KEMRI) Scientific and Ethics Review Unit (KEMRI/SERU/CVR/007–2023/4999)

### Study site

Urban areas were selected in specific cities and towns along the Northern transport corridor in proximity to the railway stations and commercial routes (highways) where container trucks make overnight stopovers (Fig 1). These areas are at risk

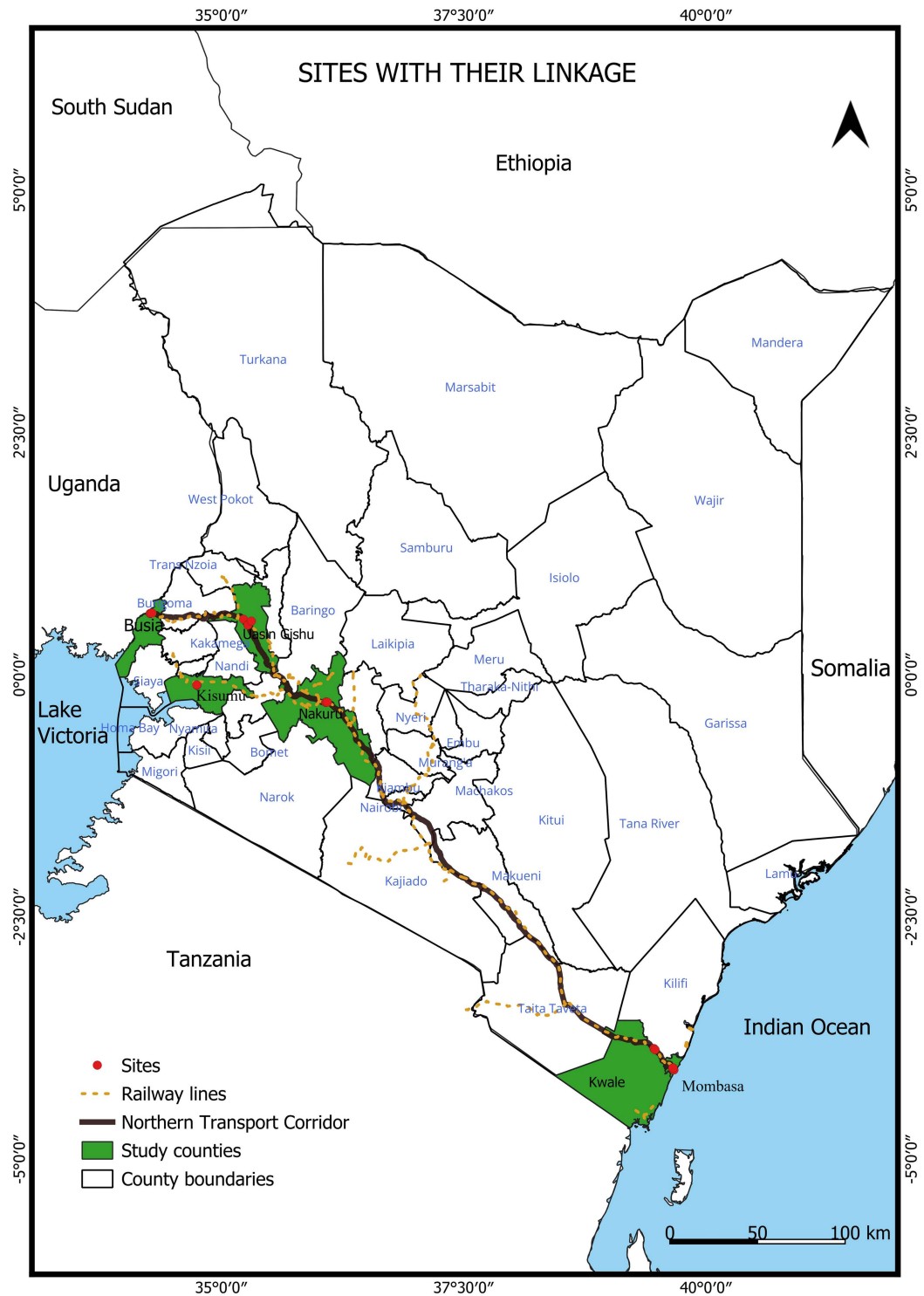

**Fig 1. Map showing the entomological sampling sites in towns and cities along the Northern transport corridor in Kenya.** The map was designed using ArcMap 10.2.2 with the ocean and lakes base layer derived from Natural Earth, a free GIS data source (naturalearthdata). The locations were collected using a GPS gadget (garmin etrex 20, https://buy.garmin.com/en-US/US/p/518046), and the county boundaries for Kenya derived from Africa Open data. (https://africaopendata.org/dataset/kenya-counties-shapefile).

of importation of pathogens from epidemic areas in the coastal region and vector invasion via human-mediated passive transport. The urban areas included coastal Mombasa (Mama Ngina grounds and the Kenya Ports Authority –KPA– precincts) and Mariakani, and mainland Nakuru, Kisumu, Busia, Malaba and Eldoret City, designated as western regions. The four cities (Mombasa, Nakuru, Eldoret and Kisumu) experience high traffic because of increased local and international travel. Mariakani and Malaba are layover sites for cargo trains and road trucks.

**Mosquito collection and identification.**

Adult *Ae. aegypti* were collected using BG-Sentinel (BGS) traps (Biogent) baited with 500g $CO_2$ (dry ice) from different study sites and identified using published morphological keys [32–34]. To avoid analyzing siblings, mosquitoes were collected from multiple sites across the sampling area. Ten BG sentinel traps were set up (6:00AM -12:00PM) outdoors and in the peri-domestic area at different locations within the study sites per day, during the long rain season. Mosquitoes were retrieved from the traps twice daily, in the morning and evening, and transported to a temporary site laboratory. The collection bags containing the trapped mosquitoes were carefully packed into biohazard bags. The mosquitoes were anesthetized using triethylamine for five minutes. Once anesthetized, the *Ae. aegypti* mosquitoes were identified to species level based on morphological characteristics. One individual mosquito was genotyped per sampling site. The use of multiple sampling sites and traps were sufficient to minimize sampling of siblings.

**Mosquito genotyping**

**DNA extraction.** Genomic DNA was extracted from 11-12 individual females' per population using Qiagen DNeasy blood/tissue kit as per the manufacturer's protocol with an additional step of adding 4ul of RNAase A to each sample to eliminate RNA. DNA was eluted to a final volume of 200 µl of 1/10X TE buffer. Samples were concentrated and normalized to 12ng/ul using Amicon Ultra 0.5uL - 30K centrifugal units prior to genotyping.

**Single Nucleotide Polymorphisms (SNPs) genotyping**

For population structure and genetic diversity analysis, we generated a SNP data set with populations of *Ae. aegypti* from throughout its global distribution, with a focus on the Kenyan population (summarized in S1 Table). Genotyping was done with the *Ae. aegypti* Axiom_aegypti1 SNP chip [31] (Life Technologies Corporation CAT#550481) [35] at the University of North Carolina Functional Genomics Core, Chapel Hill. This genotyping array contains 50,000 SNPs and was developed using representative Aedes aegypti populations worldwide to effectively capture the species' genetic diversity. It also enables the identification of *Ae. aegypti* subspecies (*Ae. aegypti aegypti* or *Ae. aegypti formosus*) [35, 36] and has been recently validated against whole-genome data [37]. Additional data from previously described samples from populations of *Ae. aegypti* collected worldwide were used as a reference panel (S2 Table). Data files were processed at Yale University with the Axiom Analysis Suite v.3.1. (Affymetrix, Santa Clara, CA) to retain those with high- quality samples (sample's dish quality control (DQC) ≥ 0.82, sample's QC call rate (QC) ≥ 66, Average call rate for passing samples (ACRPS) ≥ 90) and to call the genotypes. We genotyped 11–12 individuals *Ae. aegypti* mosquito per population at 50,000 single-nucleotide polymorphisms using the high-throughput genotyping chip, Axiom_aegypti1. The resulting dataset was complemented with previously described samples from several populations of *Ae. aegypti* collected worldwide [22, 38]. From the Kenyan and global datasets, loci that fail to genotype at 80% or more and individuals with high levels of missing data were excluded of the individuals from the global dataset were filtered out from the loci obtained from the SNP-chip that met Mendelian expectations, using the – geno 0.2 option in plink 1.9 [39]. Additionally, loci in linkage disequilibrium and those with a minor allele frequency (MAF) below 0.01 were removed (indep-pairwise 50 10 0.3, maf 0.01). Out of the 50,000 SNPs screened with the chip, 22,849 met Mendelian expectations according to Evans et al 2015 [35] and were retained. The SNP data were then filtered in PLINK v 1.9 [40, 41] to eliminate missing data at those specific locations on

the genome that did not genotype well. We also removed variants that were missing in more than 20% of individuals by specifying flag --geno 0.2. Subsequently, individuals missing more than 5% of the remaining SNPs were removed with the –mind 0.05 option.

## Admixture analyses

Geographic population structure was evaluated via Maximum Likelihood estimation of individual ancestries using the method implemented by the software Admixture 1.3.0 [42] which identifies genetic clusters and assigns proportions of each individual's ancestry to these clusters. The most likely number of clusters (K) was determined by conducting 10 independent runs from K = 1–10. We performed 10 independent runs for each K and results were summarized and plotted using the online version of CLUMPAK [43] and DISTRUCT v.1.1 [44]. The optimal K was chosen by cross validating the Q values of the ten runs using the online software Structure Harvester. The Q value with the least logarithm was considered by looking at the least value on the scatter plot [45]. To complement the admixture analysis, we performed principal component analysis (PCA) using the R ggplot2 package in R studio [46, 47]. PCA is a statistical method of data visualization commonly used in population genetics to identify structure in the distribution of genetic variation across geographical locations. The result provides a framework for interpreting PCA projections in terms of underlying processes, including migration, geographical isolation, and admixture. For the PCA analysis SNP vector data file was performed using PLINK 1.9 [40] –pca command and plotted in R studio R Core [47].

## Pearson correlation test

To assess whether *Aaa* ancestry increased with distance from the coast, *Aaa* ancestry proportions were plotted against each population distance from Mombasa and analyzed using a linear regression using R studio [44, 45] and the correlation coefficient was calculated. The genetic correlation between *Aaa* ancestry proportions—estimated using admixture at K = 2 from the global dataset—and the distance of each population from Mombasa was analyzed to assess *Aaa* introgression levels along a coastal-to-inland gradient, from Mombasa to western cities in mainland Kenya. Linear correlation analysis was performed to assess the statistical relationship between *Aaa* ancestry and distance from Mombasa (Km) using Pearson's correlation coefficient test.

## Genetic diversity

Average observed (Ho) and expected (He) heterozygosities were estimated from the SNPs dataset that included all populations from Kenya in GenoDive 3.04 [48, 49]. Genetic diversity and differentiation, pairwise genetic distances ($F_{ST}$) between all pairs of populations and their significance (significance level of 0.05) were calculated in GenoDive, using 1,000 permutations. The Genepop file generated in PGDSpider [50] was used in GenoDive to determine pairwise differentiation and genetic diversity [51]. The partitioning of the genomic variation among and within populations was evaluated through a hierarchical analysis of molecular variance( AMOVA) [52], as implemented in Genodive, using 1,000 permutations. $F_{ST}$ significance was tested using 999 permutations; standard deviations on F-statistics were obtained through bootstrapping over loci, and 95% confidence interval of F-statistics were obtained through bootstrapping over loci.

## Isolation by distance

To assess the significance of the correlation between geographic and genetic distance matrices (isolation by distance; IBD), for all Kenyan populations, we performed a Mantel test with 999 permutations in "ade4" package [53] within R v.3.4.4 [46]. The geographic distance matrix was produced from geographic coordinates in the Geographic distance matrix generator v. 1.2.3 [54]. The correlation between geographic and genetic distance was plotted in R v.3.4.4 [46] and the correlation coefficient was calculated.

### Phylogenetic analysis with the SNP data set

We evaluated the evolutionary relationships of Kenyan populations of *Ae. aegypti*, in the global context using maximum-likelihood inference on the population-level SNP data set with ascertainment bias correction for invariant sites, as implemented in IQ-Tree [55, 56]. We allowed IQ-Tree to select the best-fitting substitution model, and assessed clade support using ultrafast bootstrap values.

## Results

### SNPs quality control and validation

After quality filtering, 22,849 of the 50,000 genotyped SNPs remained. Genotype calls were generated using Affymetrix Axiom Analysis Suite 3.1 (Life Technologies–Thermo Fisher Scientific) for both newly genotyped samples and previously studied populations. This software assigns genotypes based on probe fluorescence values while excluding probes (and their corresponding SNPs) with fluorescence deviations that may indicate probe misbinding. For probe quality control and SNP calling, we primarily used default software settings in Axiom Analysis Suite to filter out genotype calls from SNPs potentially affected by off-target variants, such as those arising from insertions, deletions, or mutations at probe sites.

### Population structure

A total of 67 *Ae. aegypti* mosquitoes were genotyped; Mombasa (N = 12), Mariakani (N = 11), Nakuru (N = 11), Eldoret (N = 11), Kisumu (N = 11), and Malaba (N = 11) (for details on metadata see S1 Table). These data were analyzed together with data available from previous studies used in other publications [22, 38]. The Evanno method [57] identified run K = 2 as the optimal number of clusters. However, clustering among six *Ae. aegypti* populations were best described by K = 3. The individual admixture analysis on the Kenyan dataset (Fig 2i, K = 2) showed that the populations belong to two genetic clusters. The first cluster included samples collected from various sites in the western region (Nakuru, Eldoret, Kisumu and Malaba). The second cluster primarily included samples from coastal region (Mombasa and Mariakani). At K = 3, the data showed a third cluster consisting of samples collected from Nakuru and Eldoret. The *Ae. aegypti* populations from Mombasa in the coastal region showed a homogeneous genotype compared to other populations (Fig 2ii) for the K = 2.

The Admixture analysis of *Ae. aegypti* at the global scale (summarized in S2 Table), including historical samples (Fig 3) indicate that there is significant admixture in *Ae. aegypti* populations from the coastal region (Fig 3A K = 2). The Evanno method [57] identified run K = 2 as the optimal number of clusters. The coastal population displays admixture with *Aaa* populations. Focusing on the continental African dataset (Fig 3A and 3B), western Kenya populations cluster together with Gabon, Cameroon and Angola populations that are of the *Aaf* subspecies of *Ae. aegypti*.

The Kenyan *Ae. aegypti* populations form two genetically distinct clusters as observed in PCA, the coastal populations clusters on the right side of the plot, while Western population cluster to the left (Fig 4A). Consistent with global genome-wide SNP data, coastal populations overlap with the *Aaa* populations outside Africa, and are closer to Asian populations. PCA analysis of SNP genotypes on global data set (Fig 4B) shows that Kenyan populations are spread into three genetic clusters. One clusters with Cameroon and Gabon (Nakuru, Eldoret, Kisumu and Malaba), the other with South Africa (Mariakani), and the third one with Asia (Mombasa and Mariakani) (Fig 4B). The PCA confirms the results obtained from Admixture analysis. The result indicate that Kenya is not a pure *Aaf* population and shows a significant admixture between the Asian *Aaa* and the African *Aaf* in the coast.

Pearson correlation test for the *Aaa* ancestry with the Kenyan population shows a correlation coefficient (R²) of 0.69 which indicates a strong negative correlation (Fig 5). There is a negative correlation between the ancestry of *Ae. aegypti* and the distance from Mombasa. There is a negative correlation (Fig 5; R² = 0.69 and p-value of 2.89e-18) between the ancestry of Ae. aegypti and the distance from Mombasa among the Kenyan populations. As the distance from Mombasa increases, the ancestry proportion of *Aaa* decreases.

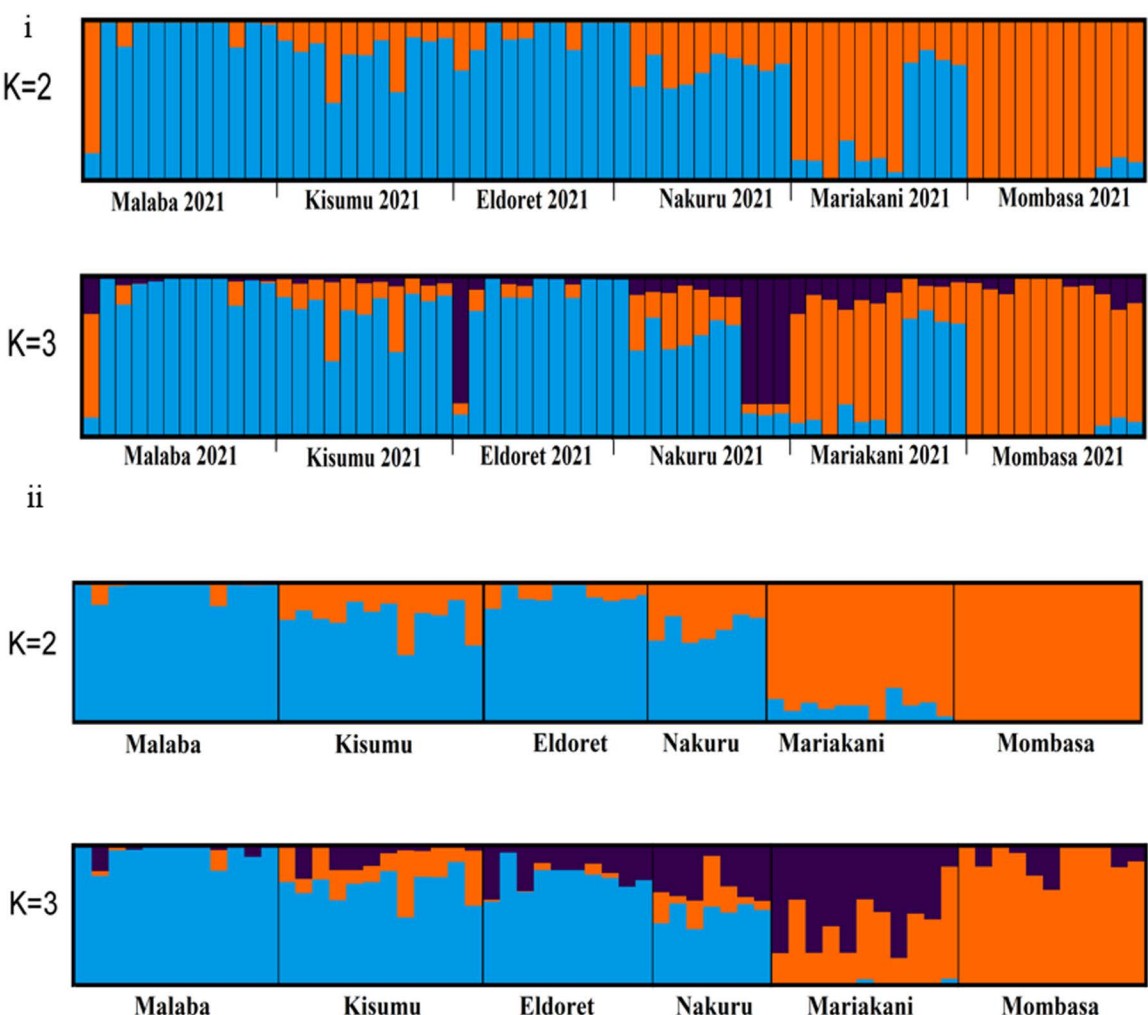

**Fig 2. Admixture bar plots for individual *Ae. aegypti* from Kenya (Fig 2i).** Population names are reported on the *x*-axis. The *y*-axis reports the probability of assignment from each individual (*Q*-value) to one of the genetic groups identified, which are represented by different colors. Each vertical bar represents an individual. Individuals with 100% assignment to one group are identified by a single color. Bars with different percentages of colors represent individuals with mixed ancestry; the thick black lines within the plots indicate population limits. **Fig 2ii**: Admixture bar plots for Kenyan *Ae. aegypti* population.

## Genetic diversity and differentiation

The pairwise estimates of genetic differentiation among the six geographical populations ranged from 0.014 between Malaba and Eldoret to 0.207 between Mombasa and Malaba (Table 1). Consistent with Admixture results, Mombasa and Mariakani form a closely related group with $F_{ST} < 0.05$. The western region showed no differentiation within, the closest population pair with Mariakani population was Kisumu ($F_{ST} = 0.088$). Pairwise $F_{ST}$ values among these study populations indicated that coastal populations had lower mean $F_{ST}$ values compared to western populations. The population pairs Mombasa-Malaba had the highest $F_{ST}$ values (0.207) observed followed by Mombasa – Eldoret. There was a lower degree of genetic differentiation among coastal and western populations of Kenya. Mean genetic diversity estimates over loci for each mosquito population are summarized (Table 2).

Analysis of molecular variation (AMOVA) indicated significant molecular variation within the individual populations compared to among populations. Genetic variation within individual population was 82%, 11% among individuals and 7%

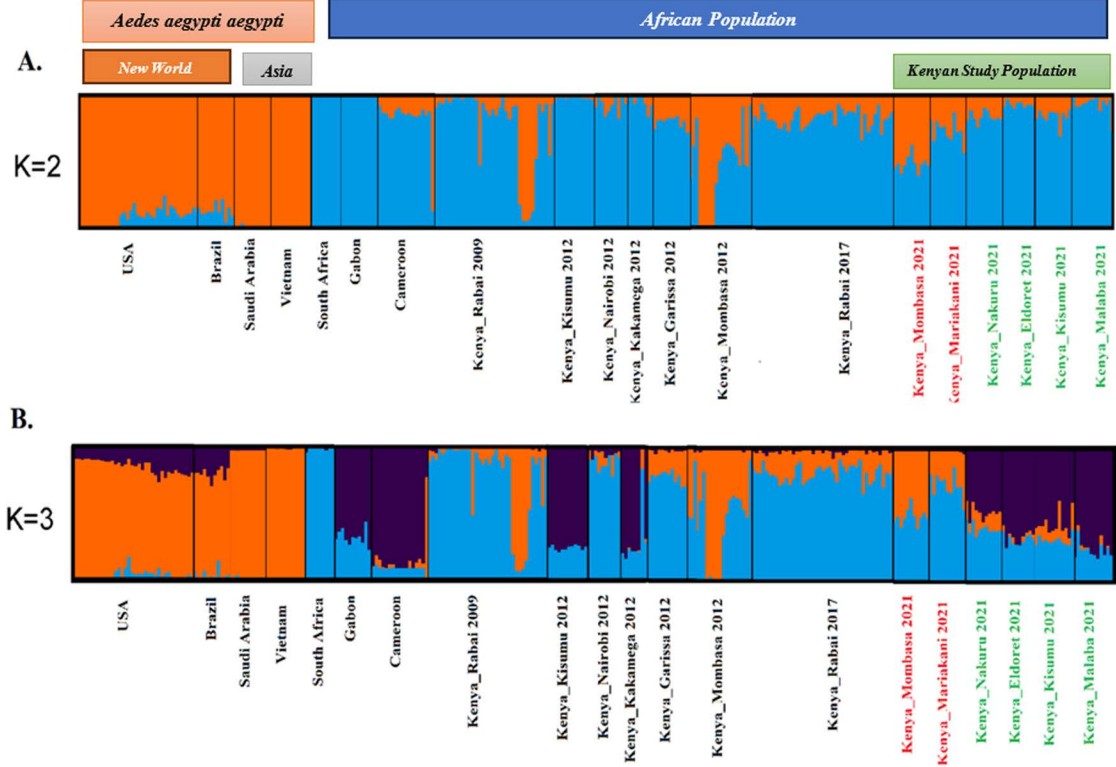

**Fig 3. Global Admixture bar plots for world *Ae. aegypti* population and Kenyan study population.** Population names are reported on the x-axis. The y-axis reports the probability of assignment from each individual (Q-value) to one of the genetic groups identified, which are represented by different colors. Each vertical bar represents an individual. Individuals with 100% assignment to one group are identified by a single color. Bars with different percentages of colors represent individuals with mixed ancestry; the thick black lines within the plots indicate population limits.

among populations (Table 3). AMOVA on the dataset indicates that most of the variation can be explained at the individual level, with a lower contribution from the population level.

## Isolation by distance

Genetic distance ($F_{ST}$) of all six *Ae. aegypti* populations in Kenya was positively correlated with geographic distance (Observation = 0.869, p = 0.0023), indicating isolation by distance (IBD; Fig 6). The genetic differentiation between *Ae. aegypti* populations showed a clear signature of IBD.

## Phylogenetic analysis

We evaluated the relationships among populations of *Ae. aegypti*, as well as other populations of the species group (S2 Table) using maximum-likelihood inference on the population-level SNP data set. The phylogeny constructed from the SNP data shows that mosquitoes from western Kenya formed a clade within the Africa population (Fig 7). All populations outside Africa form a monophyletic group distinct from all the African populations. Consistent with their admixture patterns, coastal populations are placed in an intermediate position between the *Aaa* lineage and the other Kenyan and African populations. The Nakuru, Eldoret, Kisumu and Malaba populations cluster together with the *Aaf* from Africa.

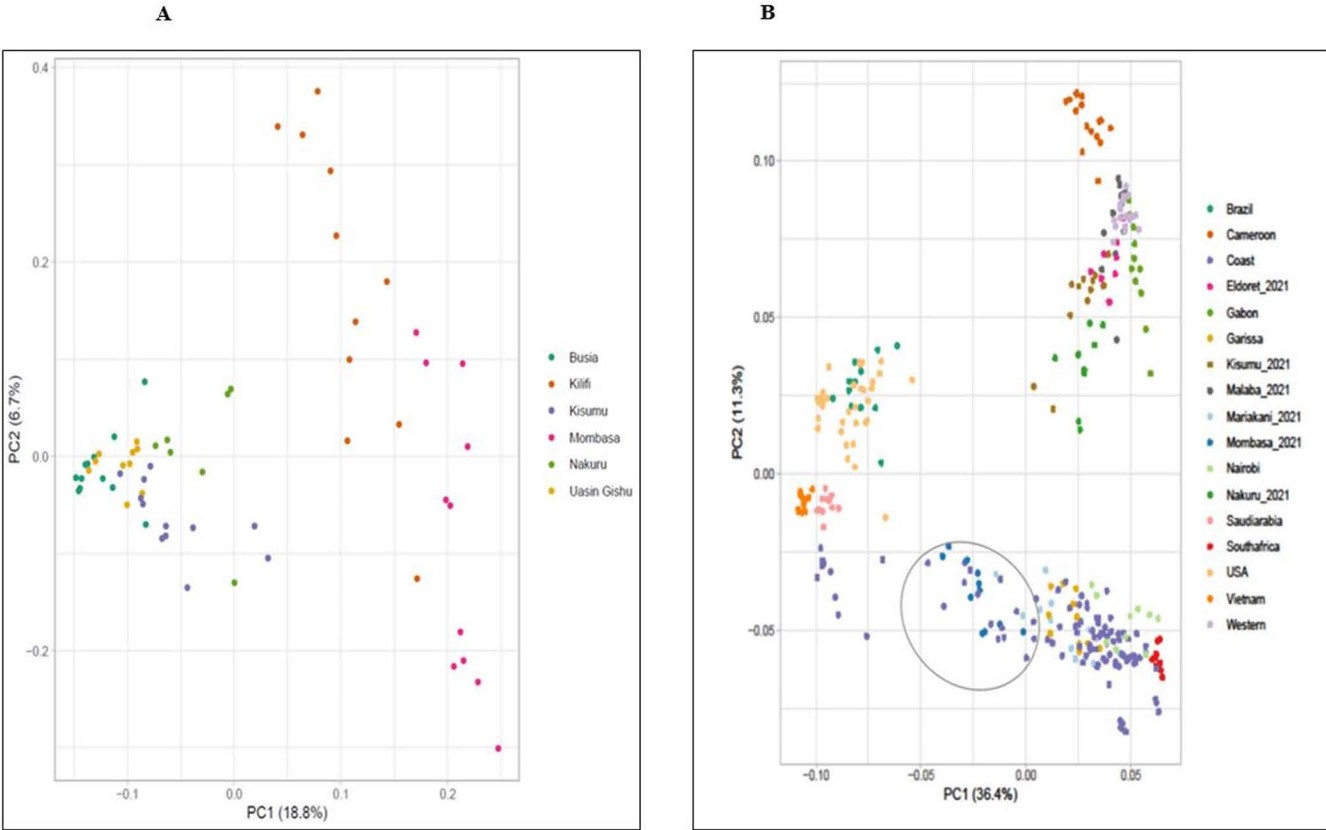

**Fig 4. Principal component analysis (PCA) on: A) the Kenyan *Ae. aegypti* populations and B) the broad global dataset.** PCA was implemented and plotted in R. ggplot package, presenting the projection of the first two PCs. Populations originating from different regions are presented with different colors as shown in the inset.

## Discussion

The *Ae. aegypti* mosquito is widely distributed throughout tropical and subtropical regions of the world. Global trade and international travel, as well as climate change, have considerably accelerated the spread of these vectors to new countries thus elevating the risk of disease transmission. This is evident in the analysis of Kenyan *Ae. aegypti* populations that shows evidence of gene flow of the invasive *Aaa* subspecies into Kenyan native *Aaf*. This invasion raises the possibility of disease epidemics in areas where they are not endemic, thus emphasizing the importance of population structure and genetic studies to identify regional genetic variants in mosquito populations. Studies have shown that the globally invasive *Aedes aegypti* populations have enhanced transmission of the majority of emerging mosquito-borne viruses such as Zika and dengue [58].

Population genetics analyses of the Kenyan *Ae. aegypti* using SNPs genetic markers (~, 22,849 SNPs) revealed two genetically distinct groups within the study populations, one in coastal and the others in western regions (Fig 2i). The western region is genetically homogenous and clusters with native African subspecies, with minimal presence of admixture between the two subspecies. The observed admixture signature in the coast is likely the result of human-mediated passive introductions to the coastal region via the shipping cargo ships from other continents. Passive human-mediated dispersal is the main cause of invasion of vectors as observed in other studies [59, 60]. The Kenyan coastal populations are genetically close to *Aaa* outside Africa, suggesting gene flow and invasion driven by long dispersal event that led

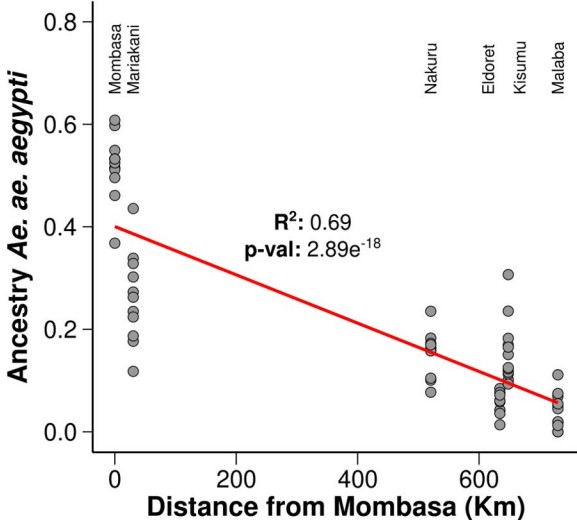

**Fig 5. Correlation test between the *Aaa* ancestry and the Kenyan populations.** X-axis: Distance from Mombasa (in kilometers), Y-axis: Ancestry proportion of *Ae. aegypti aegypti*. Gray circles represent the ancestry data of *Ae. aegypti* from different locations in Kenya. The red line is the best-fit regression line showing the trend. $R^2$ value: 0.69, indicating that 69% of the variance in ancestry. P-value: 2.89e-18, indicating a highly significant correlation.

**Table 1. Pairwise $F_{ST}$ values between the Kenyan populations.** All pairwise $F_{ST}$ estimations were significant at significance level $p < 0.005$ (S3 Table).

|  | **Mariakani** | **Malaba** | **Nakuru** | **Eldoret** | **Mombasa** | **Kisumu** |
|---|---|---|---|---|---|---|
| Mariakani | 0 | 0.138 | 0.084 | 0.117 | 0.049 | 0.088 |
| Malaba | 0.138 | 0 | 0.06 | 0.014 | 0.207 | 0.028 |
| Nakuru | 0.084 | 0.06 | 0 | 0.054 | 0.132 | 0.038 |
| Eldoret | 0.117 | 0.014 | 0.054 | 0 | 0.183 | 0.019 |
| Mombasa | 0.049 | 0.207 | 0.132 | 0.183 | 0 | 0.131 |
| Kisumu | 0.088 | 0.028 | 0.038 | 0.019 | 0.131 | 0 |

**Table 2. Summary of the mean genetic diversity indices over loci for each of *Aedes aegypti* mosquito populations.**

| Region | Population | Num | Eff-Num | Ho | Hs | Gis |
|---|---|---|---|---|---|---|
| Coastal Kenya | Mombasa | 1.807 | 1.448 | 0.243 | 0.28 | 0.131 |
|  | Mariakani | 1.806 | 1.409 | 0.225 | 0.262 | 0.143 |
| Western Kenya | Nakuru | 1.749 | 1.385 | 0.224 | 0.245 | 0.086 |
|  | Eldoret | 1.715 | 1.358 | 0.205 | 0.231 | 0.114 |
|  | Kisumu | 1.797 | 1.393 | 0.217 | 0.252 | 0.136 |
|  | Malaba | 1.727 | 1.351 | 0.197 | 0.224 | 0.121 |

Num, Number of alleles, Eff-Num, Effective number of alleles, Ho, Observed Heterozygosity, Hs, Heterozygosity Within Populations, Gis, Inbreeding coefficient

to establishment of foreign subspecies locally. Our results agree with previous studies showing possible hybrids in the coastal populations of Kenya most likely resulting from the introduction of *Aaa* from the Asian continent [22, 61].

The presence of the two subspecies in urban rural areas supports prior studies based on microsatellites that evaluated the spatial distribution of *Ae. aegypti* subspecies in Kenya [62, 63]. In this study, we collected *Aaf* from urban areas in

**Table 3. AMOVA of genetic variation in *Ae. aegypti* samples collected from different sites in Kenya.**

| Source of variation | %Var | F-stat | F-value | Std.Dev | c.i.2.5% | c.i.97.5% | P-value | F'-value |
|---|---|---|---|---|---|---|---|---|
| **Within individuals** | 0.816 | F_it | 0.184 | 0.002 | 0.18 | 0.187 | | |
| **Among individuals** | 0.111 | F_is | 0.12 | 0.002 | 0.116 | 0.123 | 0.001 | |
| **Among populations** | 0.073 | F_st | 0.073 | 0.001 | 0.071 | 0.074 | 0.001 | 0.097 |

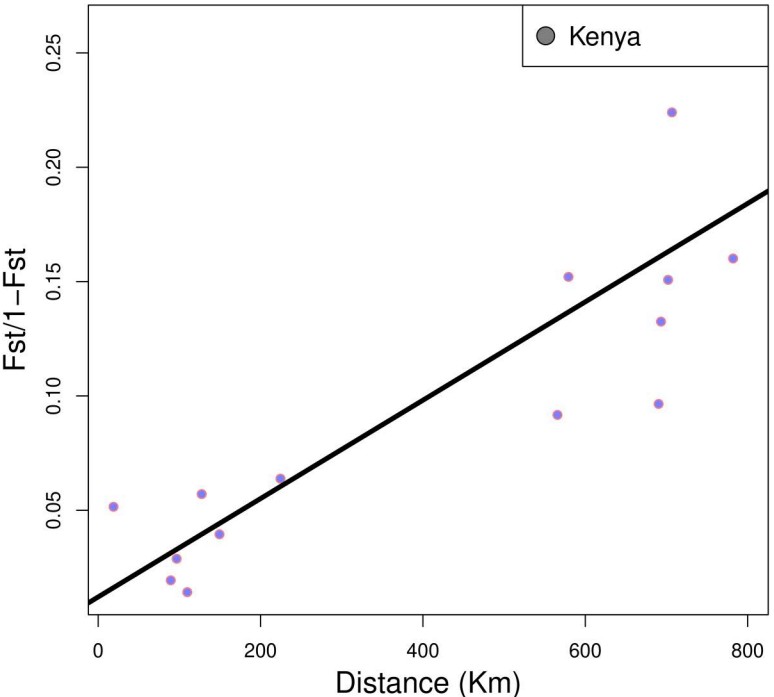

**Fig 6. Isolation-by-distance plots for all pairs of populations from Kenya.** Statistical significance was evaluated through a Mantel test as implemented in the "ade4" R package yielding a significant positive slope. The original value of the correlation between the two matrices (geographic distance and genetic distance) represented by dot. Genetic distance is given as the linearized $F_{ST}$ and geographic distance is provided in kilometers (Km).

Western Kenya (Nakuru, Eldoret, Kisumu and Malaba) and a few in Kilifi in coastal region. *Aedes aegypti* populations in Africa are known to historically breed almost entirely in forests. Today, the vector in Africa can be found in urban habitats as observed in previous studies [22, 64, 65]. We observed the foreign genetic signature at high frequency in the Kenyan coastal region, with frequency decreasing as distance from the sea to the mainland, in agreement with the previous studies [63]. Western Kenya had lower presence of *Aaa* ancestry. The presence of native population in urban/cities is the result of increased urbanization, with human settlements encroaching into forests. This observation concurs with previous studies that observed high abundance of the vector in urban areas [7, 9]. Such an increase in urbanization in African urban areas could confer a greater adaptive advantage for *Aaa* over *Aaf* since it is adapted to urban environments.

The observed clustering of coastal populations with Asian population samples is likely the result of historical trade between the two continents. The slave trade likely led to the introduction of the foreign *Aedes* populations to the East African coast in the early 1900s which is now interbreeding with the native population [20, 29]. Admixture may complicate efforts to control *Ae. aegypti* by shifting typical behaviors associated with this mosquito in a way that may allow them to evade control measures that used to be effective against them [66]. The Kenyan coastal region had sporadic outbreaks

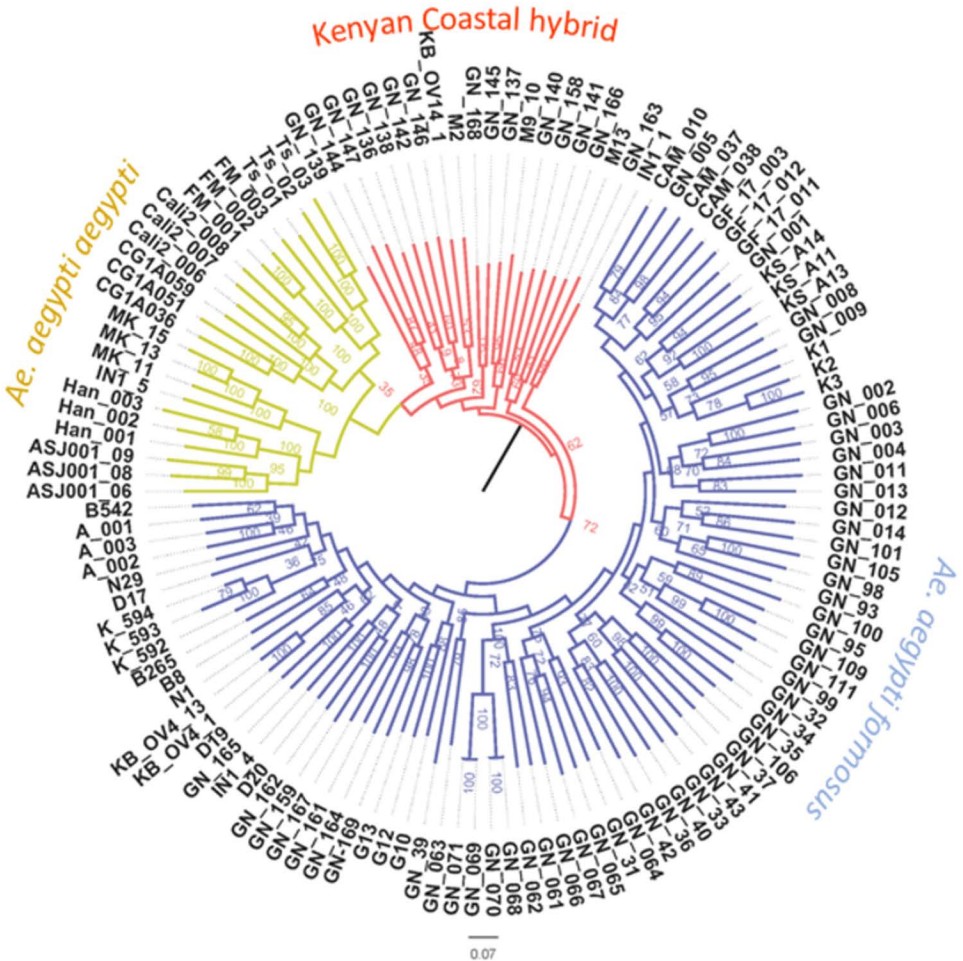

**Fig 7. Maximum likelihood (ML) annotated phylogenetic tree re-constructed using a panel of ~22,849 SNPs.** *Aaa* samples from New World and Asia were used to test the distinctiveness of *Aaf* and *Aaa* lineages (for details on branch/node labels see S2 Table).

of dengue and chikungunya virus in the past but is now becoming endemic for both emerging diseases; this may be attributed to the high human bite-rate of the hybrid species of *Ae. aegypti* observed in the coastal region [67], and its demonstrated high competence compared to native *Aaf* (Aubry 2020). Prior studies of Kenyan outbreaks indicated that the viruses could have originated in South East Asia [68, 69]. This was also evident in our study which showed that the Asian populations are a likely source of *Aaa* invasion into Kenya, possibility though importation of infected mosquitoes/ eggs. The availability of highly competent vectors has amplified arbovirus outbreaks in the Kenyan coastal region. The detection of dengue and chikungunya in sero-prevalence studies in mainland cities and towns is an indicator that the infections are spreading to the mainland as evident in parts of western Kenya [70, 71]. This low transmission is likely due to the presence of less efficient subspecies of *Aaf* vectors observed in those regions [58].

Genetic patterns observed in the study suggest that the coastal *Ae. aegypti* populations in study cities are admixed. This is in contrast with previous studies where the presence of the two subspecies coexisting sympatrically in peri-domestic and sylvan populations in coastal Kenya was observed [29]. The distribution of the two subspecies of *Ae. agypti* in Rabai in coastal region was observed where the species were closely breeding in the same ecosystem. Historical literature from the 1960s suggests that the two subspecies remained distinct in Rabai [4] for some time but formed mixed hybrid swarms

along the coast. This is highly consistent with our findings. It is not clear why the two coexisted for so long in the Rabai environment without mixing, but mixed more freely just tens of kilometers away. Up-to-date the 'pure' *Aaa* have not been found along the coast since around 2010 [72]. The present study highlights the complex patterns of vector colonization that have occurred in Kenya. Previous studies speculated that the presence of *Aaf* in urban areas rather than in rural/forest areas which are typical for the species, was due to the increased urbanization and human activities such as deforestation [73]. The observed admixed population in the coastal region, most likely reflects the historical relationship of the Kenyan coastal region with the Asian and European region during the slave trade and continuous trade between the two continents. The Arab and Asian traders used the coastal region as a port-of-call for ship provisioning during travels between Europe and the African continent and also as a slave outpost for the Atlantic slave trade [74]. The considerable movement of ships/trade between the islands and both continents could have provided ample opportunity for the introduction of the foreign mosquito species. Mombasa is a tourist destination area allowing influx of tourists from all over the world, some from endemic areas for known viruses; these may potentially introduce viruses that are known to be transmitted by the *Ae. aegypti*. The introduction of these pathogens may place the native communities in danger due to observed high hybridization between the two subspecies that may promote a more opportunistic feeding behavior thus increasing the chance for the accidental transmission of viruses to humans.

## Conclusion

Human economic activity is the primary cause of *Ae. aegypti* invasion into foreign countries as observed by the presence of *Aaa* ancestry in mosquito populations from coastal Kenya. Human-mediated transportation and migration are facilitating long distance vector dispersal and could drive admixture of forest-adapted and urban-adapted populations leading to increased adaptive flexibility, with implications for disease transmission and control. Effective, efficient, and sustainable control of invasive mosquitoes, such as *Ae. Aegypti* and identification of the routes of introduction and dispersal will be key to prevent future incursions and assess their potential health threat.

## Supporting information

**S1 Table.  Kenyan *Aedes aegypti* populations.**
(DOCX)

**S2 Table.  Admixture analysis of *Ae. aegypti* at the global data set. *Aedes aegypti* world and Kenyan populations included in the phylogenetic analysis** .
(XLSX)

**S3 Table.  Pairwise FST significant values for the Kenyan populations.**
(DOCX)

## Acknowledgments

We recognize the technical support of Viral Hemorrhagic Fever Lab team members (KEMRI, Nairobi). We are grateful to fuel Owaka, GIS support unit, KEMRI for producing the map of the study area. We are also grateful for the support from the county staff, local chiefs as well as community members of Mombasa, Kilifi, Nakuru, Eldoret, Kisumu and Busia Counties.

## Author contributions

**Conceptualization:** Francis Mulwa, Rosemary Sang, Andrea Gloria-Soria, Joel Lutomiah.

**Data curation:** Francis Mulwa, Solomon Langat, Noah Rose, Armanda Bastos, Andrea Gloria-Soria, Joel Lutomiah.

**Formal analysis:** Francis Mulwa, Dario Balcazar, Solomon Langat, James Mutisya, Betty Chelangat, Carolyn. S McBride, Noah Rose, Jeffrey Powell, Armanda Bastos, Andrea Gloria-Soria, Joel Lutomiah.

**Funding acquisition:** Jeffrey Powell, Rosemary Sang.

**Investigation:** Francis Mulwa, Armanda Bastos, Joel Lutomiah.

**Methodology:** Francis Mulwa, Dario Balcazar, Carolyn. S McBride, Noah Rose, Jeffrey Powell, Rosemary Sang, Armanda Bastos, Andrea Gloria-Soria, Joel Lutomiah.

**Project administration:** Joel Lutomiah.

**Resources:** Carolyn. S McBride, Jeffrey Powell, Rosemary Sang, Joel Lutomiah.

**Supervision:** Dario Balcazar, Jeffrey Powell, Rosemary Sang, Armanda Bastos, Andrea Gloria-Soria, Joel Lutomiah.

**Visualization:** Francis Mulwa, Rosemary Sang, Armanda Bastos.

**Writing – original draft:** Francis Mulwa.

**Writing – review & editing:** Francis Mulwa, Dario Balcazar, Solomon Langat, James Mutisya, Betty Chelangat, Carolyn. S McBride, Noah Rose, Jeffrey Powell, Rosemary Sang, Armanda Bastos, Andrea Gloria-Soria, Joel Lutomiah.

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
