## [Decision Letter · Decision Letter 0]

30 Jan 2025

Response to Reviewers
Revised Manuscript with Track Changes
Manuscript

Shaden Kamhawi

co-Editor-in-Chief

Paul Brindley

co-Editor-in-Chief

**Journal Requirements:**

At this stage, the following Authors/Authors require contributions: Francis Mulwa Musili. Please ensure that the full contributions of each author are acknowledged in the "Add/Edit/Remove Authors" section of our submission form.

Potential Copyright Issues:

- Figure 1; Please provide a direct link to the base layer of the map (i.e., the country or region border shape) and ensure this is also included in the figure legend; and provide a link to the terms of use / license information for the base layer image or shapefile. We cannot publish proprietary or copyrighted maps (e.g. Google Maps, Mapquest) and the terms of use for your map base layer must be compatible with our CC BY 4.0 license.

5) We note that your Data Availability Statement is currently as follows: "I do accept the data to be available to the public". Please confirm at this time whether or not your submission contains all raw data required to replicate the results of your study. Authors must share the “minimal data set” for their submission. PLOS defines the minimal data set to consist of the data required to replicate all study findings reported in the article, as well as related metadata and methods (https://journals.plos.org/plosone/s/data-availability#loc-minimal-data-set-definition).

- The points extracted from images for analysis..

**Reviewers' comments:**

**Key Review Criteria Required for Acceptance?**

**Methods**

-Are the objectives of the study clearly articulated with a clear testable hypothesis stated?

-Is the study design appropriate to address the stated objectives?

-Is the population clearly described and appropriate for the hypothesis being tested?

-Is the sample size sufficient to ensure adequate power to address the hypothesis being tested?

-Were correct statistical analysis used to support conclusions?

-Are there concerns about ethical or regulatory requirements being met?

Reviewer #1: The objectives of the study which, are to Investigate the population structure of Ae. aegypti in Kenya, to determine the extent of admixture between Ae. aegypti aegypti (Aaa) and Ae. aegypti formosus (Aaf) in Kenya, and to determine the global phylogenetic clustering of Ae. aegypti populations are well articulated. The study design, which employs the use of SNPs to determine genetic structure, diversity, and phylogenetics is satisfactory. The Aaa and Aaf populations are well placed to test the study hypothesis. The sample size of each region is lower than the expected 30 samples per population. However, if none of the analysis used an assumed number of populations, the sample size should e OK. The criteria used for sample size determination for each population. The statistical analyses and population genetics analyses used were sufficient, though there should be a clear distinction in the methodology explanation between the correlation test and isolation by distance. The study sought KEMRI Ethical clearance which is nationally acceptable.

Reviewer #2: The objective of this study is clear and archivable , but the methodology was not rigorous with a gap in the description. Unfortunatly, author did not describe data quality control and validation section ,making it difficult to validate the results presented.

Reviewer #3: Clarity of Study Objectives and Hypothesis

The objective of the study is not clearly articulated in the abstract or introduction. While the study discusses genetic mixing between populations, there is some inconsistency in the terminology used, particularly with "hybridization" and "gene flow." Clarifying whether the study focuses on hybridization, which involves introgression, or admixture, which involves gene flow, would improve the overall clarity.

Furthermore, the stated hypothesis—"based on the frequency and distribution of dengue and chikungunya epidemics, we hypothesize that coastal Ae. aegypti populations have a higher proportion of Aaa ancestry than those from other locations in Kenya"—seems problematic. The observed disease distribution is influenced by multiple factors, including urbanization, population density, and water availability, making the connection between disease distribution and genetic ancestry somewhat tenuous. Additionally, the coastal regions' genetic composition is likely shaped by historical trade and introduction of new alleles, introducing an autocorrelation issue. A broader and more robust justification for the hypothesis is recommended.

Study Design Appropriateness

The study design appears appropriate for addressing the stated objectives. The use of genetic analyses to examine ancestry proportions in different populations is a valid approach.

Description of Population

The population under study is clearly described and relevant to the hypothesis being tested. The authors provide a geographic and ecological context for the sampled mosquito populations, which supports the genetic comparisons being made.

Sample Size Considerations

The study genotyped a total of 67 Ae. aegypti mosquitoes, which is on the lower end for population genetic studies but should still provide sufficient statistical power. However, the sample size is not addressed in the abstract, which is an omission that should be corrected. Including a statement on sample size in both the abstract and the study site/mosquito collection section of the methods would enhance transparency and allow for a better assessment of the study’s robustness.

Statistical Analyses

The statistical analyses used in the study are appropriate and well-supported. Population structure was assessed using Admixture 1.3.0. Principal Component Analysis (PCA) was implemented in R. Genetic diversity was assessed using GenoDive 3.04, with heterozygosities (Ho and He) and pairwise FST values computed for population differentiation. Isolation by distance was tested with a Mantel test using the ade4 package in R, and phylogenetic relationships were inferred using IQ-Tree with ascertainment bias correction for invariant sites. These methods collectively provide a robust statistical framework for assessing population structure, gene flow, and evolutionary relationships among Ae. aegypti populations in Kenya. 

Ethical and Regulatory Considerations

There are no apparent concerns regarding ethical or regulatory requirements in this study.

Summary of Recommended Improvements

- Clarify the objective: The study’s objective should be explicitly stated in both the abstract and introduction.

- Improve terminology consistency: Clearly differentiate between "hybridization" (with introgression) and "gene flow" (admixture) to enhance clarity.

- Strengthen hypothesis justification: The hypothesis should be supported by a broader set of explanations rather than solely correlating disease frequency with ancestry proportions.

- Address sample size transparently: The sample size should be mentioned in the abstract and methods to improve study transparency.

By addressing these points, the study’s methodological clarity and overall rigor would be significantly improved.

**Results**

-Does the analysis presented match the analysis plan?

-Are the results clearly and completely presented?

-Are the figures (Tables, Images) of sufficient quality for clarity?

Reviewer #1: The analysis results correlated with what was described in the methodology section. The results are clearly presented. However, Table S1 would be clearer if the number of samples per population was summarized, rather than presenting one sample per row. Figure 4 needs higher resolution to be more legible.

Reviewer #2: Aedes aegypti population structure, genetic diversity, and isolation by distance using genome-wide

single nucleotide polymorphism (SNPs) datasets are generated and well presented , but data validation, a crucial step at both sample and SNP level has not been performed. The Author did not provide results of these validations and therefore does not assure their credibility.

Reviewer #3: Alignment with Analysis Plan

The analysis presented in the results section is consistent with the planned methods. The study follows through on the stated objectives, employing appropriate genetic analyses to assess population structure, genetic diversity, and isolation by distance. The results are well-supported by the statistical methods outlined in the methodology section.

Clarity and Completeness of Results

The results are clearly and comprehensively presented. The findings regarding the genetic structure of Ae. aegypti populations in Kenya are well-explained, with detailed descriptions of how the data support the conclusions. The observed admixture in coastal populations and the genetic homogeneity of western populations are appropriately framed within the study's objectives. Additionally, the correlation between genetic ancestry and geographic distance is effectively demonstrated.

Quality of Figures and Tables

The figures and tables included in the results section are of high quality. The graphical representations of admixture analysis, principal component analysis (PCA), phylogenetic trees, and isolation by distance plots are clear and well-labeled. The choice of colors and presentation style enhances readability, making it easy to interpret the genetic relationships and population structure.

**Conclusions**

-Are the conclusions supported by the data presented?

-Are the limitations of analysis clearly described?

-Do the authors discuss how these data can be helpful to advance our understanding of the topic under study?

-Is public health relevance addressed?

Reviewer #1: The conclusion has clearly and concisely summarized the inferences that have been made based on the results of the study. Unfortunately, the study limitations have not been highlighted clearly. The authors clearly articulate the potential implication of the genetically admixed coastal samples on mosquito-borne disease transmission and mosquito control. Mosquito-borne diseases are a big issue in public health. Therefore, understanding the genetic structure of mosquito vectors and its implication on disease transmission and vector control is of great public health importance.

Reviewer #2: Although the Homogenous population that clusters with African Ae. aegypti formosus (Aaf) is found.

The coastal mosquitoes showed evidence of admixture between the two subspecies. Aedes aegypti populations in Kenya display a positive correlation between genetic distance (FST) and geographic distance, suggesting isolation by distance.

sub-species admixture may result in higher human feeding preference, biting rates and potentially altered vectorial competence, facilitating dengue and chikungunya outbreaks. The hybridation is evidenced,these conclusion might be true, but data are not beeing validated, makes these data invalid.

Reviewer #3: Conclusion Review

Are the conclusions supported by the data presented?

Yes, generally. However, one issue is the use of the term “hybrid.”

Recommendation on the Use of “Hybrid” Terminology

The study refers to Ae. aegypti populations in coastal Kenya as “hybrids.” However, hybridization generally implies some degree of reproductive isolation or barriers to free interbreeding, which has not been explicitly demonstrated in the study. If the two subspecies are freely interbreeding without constraints, the more appropriate term might be “admixed populations.”

To strengthen their justification for using “hybrid,” the authors could:

Examine Isolation by Distance at Different Scales

If gene flow is continuous across populations, this would indicate a gradient of admixture rather than discrete hybrid populations.

Testing for IBD at multiple scales could help determine whether genetic exchange is restricted or widespread.

Explicitly test for spatial autocorrelation using Moran’s I or a related analysis to determine whether Aaa and Aaf form a gradient rather than discrete groups when accounting for the spatial clustering of the collection points.

Test for Ancestral Hybridization

Methods such as TreeMix or Approximate Bayesian Computation (ABC) could help determine whether hybridization was a historical event with subsequent free interbreeding, or if distinct genetic clusters persist.

If past hybridization followed by genetic homogenization is supported, referring to the populations as “admixed” rather than “hybrids” may be more precise.

Clarify the Definition of “Hybrid” in This Context

If the authors intend to keep the term “hybrid,” they should explicitly define what they mean and provide evidence that Aaa and Aaf maintain some level of genetic distinction despite admixture.

Alternatively, using “admixed populations” would avoid potential misinterpretation.

Are the limitations of the analysis clearly described?

Yes, generally. However, while the manuscript acknowledges that sampling sites are not evenly distributed across Kenya, with a stronger focus on coastal regions, simply stating this as a limitation is insufficient. Uneven sampling could introduce bias in detecting genetic gradients and may not fully capture the broader population dynamics across the country.

Recommendation:

The authors should address this issue analytically, as there is well-established theory and statistical methodology in landscape genetics to handle such biases. Possible approaches include:

Incorporating spatially explicit models to account for geographic sampling biases.

Using spatial regression techniques or geographically weighted analyses to ensure observed genetic patterns are not artifacts of sampling distribution.

Conducting sensitivity analyses to determine the extent to which uneven sampling may influence results.

Do the authors discuss how these data can be helpful to advance our understanding of the topic under study?

Yes. The study contributes to understanding how genetic admixture may shape mosquito populations and potentially influence vectorial capacity. The findings add valuable insights into mosquito population dynamics and the potential implications of gene flow between subspecies.

Is public health relevance addressed?

Yes. The study discusses how genetic admixture in Ae. aegypti populations may impact disease transmission dynamics, particularly for dengue and chikungunya. The implications for vector control strategies and surveillance programs are clearly articulated.

**Editorial and Data Presentation Modifications?**

Reviewer #1: Minor editorial suggestions have been appended to the main text as comments.

Reviewer #2: The whole section of data validataion is missing . An additional file for data validation should be inserted as well as the methodology used.

Reviewer #3: Edit the use of the term "hybrid" in the text, figures, and tables as appropriate following revisions.

**Summary and General Comments**

Reviewer #1: Mosquitoes are an important disease vector, especially in the Northern Kenyan corridor that has shown emergence of new arboviral diseases. This study is important to understand the population structure of Ae. aegypti in Kenya, and highlights admixture of subspecies in coastal Kenya, which can influence disease transmission and vector control. Using genome-wide SNP analysis provides a robust method of evaluating genetic structure.

The study strengths include the important public health implications of the findings, the methods used, and the novelty of these analysis which have not been reported before.

The study weakness, which can become an opportunity, is the validation of genetic traits in admixed Ae. Aegypti subspecies, such as vectoral capacity and human feeding preferences.

Reviewer #2: This paper is addressing the Population genetic analysis of the two devastating subspecies of Aedes aegypti, implicated in the transmission of dengue and chikungunya virus in Kenya. Results generated could account for the understanding of the distribution of epidemics and envisage a better management of the species and the disease control, but:

1. The author should include a clear quality check and validation methods steps , then present related results in the results section

2. Section of genetic diversity indices results should be presented in a table form, to preside the structure, PCA and phylogenetic results.

The results presented in this paper, although important are insufficient and therefore, make the decision of the paper difficult.

Reviewer #3: This study presents an interesting and relevant analysis of Aedes aegypti populations in Kenya, exploring genetic structure, admixture, and potential implications for disease transmission. The study employs robust genomic methods and presents well-visualized results that contribute valuable insights to the field. The findings are particularly significant for understanding mosquito population dynamics and the potential consequences of gene flow between subspecies.

Strengths:

Well-executed genomic analysis using appropriate tools and methods.

Clearly structured results with high-quality figures and tables.

Public health relevance is well addressed, connecting genetic patterns to potential vectorial capacity.

The study contributes to the broader discussion of how genetic admixture may influence disease transmission.

Weaknesses & Areas for Improvement:

The use of the term "hybrid" lacks strong justification; the authors should either provide additional evidence or reconsider terminology.

Sampling bias due to uneven geographic coverage is acknowledged but not sufficiently addressed analytically. The inclusion of spatially explicit modeling or sensitivity analysis would strengthen conclusions.

The manuscript does not fully explore potential ecological and environmental factors that may shape genetic structure beyond human-mediated movement.

Some interpretations, particularly regarding the connection between genetic patterns and disease outbreaks, could be framed with greater caution to avoid overgeneralization.

Novelty & Significance:

This study provides new data on the genetic structure of Ae. aegypti in Kenya, contributing to the understanding of population movement, gene flow, and potential hybridization. Given the role of Ae. aegypti in vector-borne disease transmission, this research has significant applied value for vector control and public health planning.

Publication & Research Ethics:

No concerns regarding dual publication, research ethics, or data integrity were noted.

Recommendation:

While the study is methodologically sound and contributes valuable insights, addressing the concerns related to terminology, sampling bias, and analytical rigor would enhance the manuscript’s impact. If these revisions are made, the paper will be a strong contribution to the field.

PLOS authors have the option to publish the peer review history of their article (what does this mean? ). If published, this will include your full peer review and any attached files.

**Do you want your identity to be public for this peer review?** For information about this choice, including consent withdrawal, please see our Privacy Policy .

Reviewer #1: No

Reviewer #2: **Yes: ** Abraham Mayoke

Reviewer #3: No

**Figure resubmission:****Reproducibility:** To enhance the reproducibility of your results, we recommend that authors of applicable studies deposit laboratory protocols in protocols.io, where a protocol can be assigned its own identifier (DOI) such that it can be cited independently in the future. Additionally, PLOS ONE offers an option to publish peer-reviewed clinical study protocols. Read more information on sharing protocols at https://plos.org/protocols?utm_medium=editorial-email&utm_source=authorletters&utm_campaign=protocols

---

## [Decision Letter · Decision Letter 1]

8 Apr 2025

Dear Mr Musili,

We are pleased to inform you that your manuscript 'Population genetic analysis of Aedes aegypti reveals evidence of emergingAdmixture populations in coastal Kenya' has been provisionally accepted for publication in PLOS Neglected Tropical Diseases.

Best regards,

Paul O. Mireji, PhD

Section Editor

Paul Mireji

Section Editor

Shaden Kamhawi

co-Editor-in-Chief

Paul Brindley

co-Editor-in-Chief

Reviewer's Responses to Questions

**Key Review Criteria Required for Acceptance?**

**Methods**

-Are the objectives of the study clearly articulated with a clear testable hypothesis stated?

-Is the study design appropriate to address the stated objectives?

-Is the population clearly described and appropriate for the hypothesis being tested?

-Is the sample size sufficient to ensure adequate power to address the hypothesis being tested?

-Were correct statistical analysis used to support conclusions?

-Are there concerns about ethical or regulatory requirements being met?

Reviewer #1: The objectives of the study which, are to Investigate the population structure of Ae. aegypti in Kenya,

to determine the extent of admixture between Ae. aegypti aegypti (Aaa) and Ae. aegypti formosus

(Aaf) in Kenya, and to determine the global phylogenetic clustering of Ae. aegypti populations are

well articulated. The study design, and methods used to evaluate genetic diversity and differentiation, and statistical analyses implemented thereof are sufficient.

**Results**

-Does the analysis presented match the analysis plan?

-Are the results clearly and completely presented?

-Are the figures (Tables, Images) of sufficient quality for clarity?

Reviewer #1: The revised supplementary material cannot be viewed because the documents are not hyperlinked. I require access to these files to see whether the issues raised in the original review have been addressed.

**Conclusions**

-Are the conclusions supported by the data presented?

-Are the limitations of analysis clearly described?

-Do the authors discuss how these data can be helpful to advance our understanding of the topic under study?

-Is public health relevance addressed?

Reviewer #1: The study limitations are not highlighted.

**Editorial and Data Presentation Modifications?**

Reviewer #1: The majority of editorial/grammatical issues raised in the first review have not been addressed. These comments were appended onto the annotated document.

**Summary and General Comments**

Reviewer #1: The manuscript has been made clearer with better flow and logical information from the introduction, to the methods, to results, then to discussion and conclusions.

PLOS authors have the option to publish the peer review history of their article (what does this mean? ). If published, this will include your full peer review and any attached files.

**Do you want your identity to be public for this peer review?** For information about this choice, including consent withdrawal, please see our Privacy Policy .

Reviewer #1: **Yes: ** Winnie Akoth Okeyo

---

## [Editor Report · Acceptance letter]

Dear Mr Mulwa,

We are delighted to inform you that your manuscript, "Population genetic analysis of Aedes aegypti reveals evidence of emerging Admixture populations in coastal Kenya ," has been formally accepted for publication in PLOS Neglected Tropical Diseases.

Best regards,

Shaden Kamhawi

co-Editor-in-Chief

Paul Brindley

co-Editor-in-Chief
